# Exploring Complex Trade-offs in Information Bottleneck through Multi-Objective Optimization

## Abstract

Information Bottleneck (IB) theory provides a principled approach to analyze and optimize how neural networks extract and learn latent representations from data, aiming to enhance network performance and generalization. The IB framework has been applied and validated across various domains in deep learning. However, most studies employing IB require tuning of Lagrange multipliers to balance compression and prediction during optimization. Finding the optimal Lagrange multiplier $\beta$ to achieve the best balance between compression and prediction is challenging, relying heavily on empirical tuning and potentially failing to capture the complex trade-offs present within the IB paradigm. In this paper, we redefine the IB problem as a multi-objective optimization problem with respect to compression and prediction objectives. We employ a gradient-based multi-objective optimization algorithm that adaptively determines the weights for this optimization challenge. Our method is demonstrated to automatically find Pareto-optimal solutions, achieving a balance between compression and prediction, and exploring more complex Pareto frontiers than linear weighting. We compare our approach with the Variational Information Bottleneck and its variants across different datasets. Empirical results confirm that our method achieves a more stable and optimal trade-off compared to Information Bottleneck approaches with manually-tuned multipliers. The code is available in `https://anonymous.4open.science/r/ASDGASDG`.

## 1 Introduction

Neural networks are powerful learning models that can adapt to complex patterns in data and excel at various tasks Blazek & Lin (2021); Kriegeskorte & Golan (2019); Dai (2021); Sutskever et al. (2014). How do highly parameterized neural networks exhibit good fit performance as well as generalization while memorizing datasets? One possible answer is that neural networks can learn latent representations from datasets Dabagia et al. (2022); Kilinc & Uysal (2018); Yang et al. (2022). Latent representations are features that capture fundamental and hidden information about the data, such as shapes, colors, or textures in images, or themes, sentiments, or styles in text Ye & Shen (2020); Pati & Lerch (2021); Bau et al. (2017). By learning these latent representations, neural networks can ignore irrelevant or noisy details and focus on what matters for the task Schneider et al. (2023); Dabagia et al. (2022).

A longstanding issue in deep learning is understanding the nature and quality of these latent representations and how neural networks learn them from data Hafner et al. (2019); Mo et al. (2020); Wieczorek et al. (2018). Good latent representations are those that capture the underlying features and structure of the data while discarding irrelevant and noisy details Zhang et al. (2017); Kim et al. (2020). Learning such representations is crucial for achieving high performance and generalization in various deep learning tasks, including image classification, natural language processing, and reinforcement learning Mo et al. (2020); Gelada et al. (2019); Ye & Bors (2020). However, it is not yet clear how neural networks learn these representations from data and what factors influence the quality and efficiency of the learned representations. How can neural networks be guided to learn more optimized representations? One possible answer is through the use of Information Bottleneck theory Harremoës & Tishby (2007); Tishby & Zaslavsky (2015), which provides a rigorous framework

for examining the trade-off between the complexity and utility of latent representations in terms of mutual information. The Information Bottleneck method aims to find a succinct representation of the input random variable $X$ that preserves relevant information about the target variable $Y$, given their joint probability distribution $p(X, Y)$ Geiger & Kubin (2020); Li & Liu (2021).

Despite challenges in estimating mutual information, the Information Bottleneck method has been applied with success in various deep learning tasks due to the theoretical insights of the Information Bottleneck framework and the variational techniques implemented by it Geiger & Kubin (2020); Tishby & Zaslavsky (2015), such as computer vision Voloshynovskiy et al. (2019); Su et al. (2023); Lee et al. (2021), natural image processing Zhang et al. (2022); Geiger & Kubin (2020); Mai et al. (2022), self-supervised learning of graphs Wu et al. (2020); Alemi et al. (2017); Gu et al. (2022), and compression of neural networks Dai et al. (2018); Tishby & Zaslavsky (2015). However, while these studies benefit from the IB framework Razeghi et al. (2023); Zhai & Zhang (2022); Dai et al. (2018), they also face significant **limitations**: *they follow the Information Bottleneck and VIB setup, linearly weighting two important objectives with hyperparameters and conducting extensive experiments to manually tune the parameters to achieve the desired compression-prediction trade-off, i.e.,* **_a complex and expensive manual process of finding trade-offs._** As shown in Figure 1, we visualize the information plane of the three mainstream IB methods on the MINIST dataset. It can be significantly found that the process of finding trade-offs between different methods varies greatly, but they all need complex optimization processes.

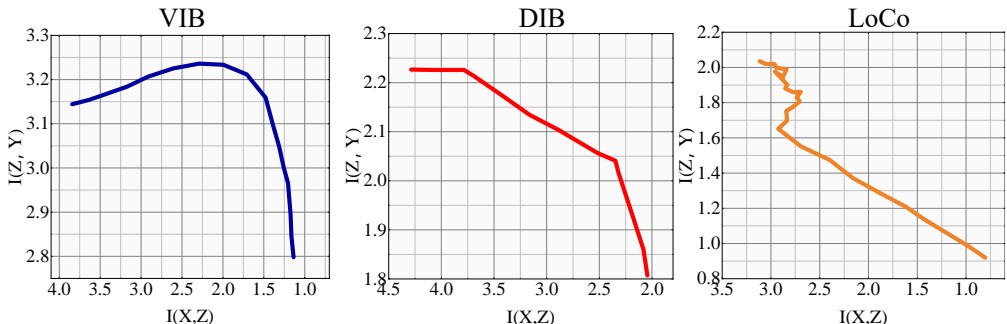

Figure 1: The Information Plane of three different IB methods, i.e., VIB Alemi et al. (2017), DIB Pan et al. (2021), and LoCoButakov et al. (2024).

To better understand the above limitations, we revisit and analyze the Information bottleneck (IB) optimization problem, and experimentally find that the choice of Lagrange multiplier significantly affects the performance and robustness of the model, as well as the mutual information between the target and the data. In addition, manually setting a plausible $\beta$ may not be the optimal solution to trade off compression and prediction goals. To this end, for the first time, we redefine this problem as a multi-objective optimization problem with respect to compression and prediction objectives from the perspective of multi-objectives. Furthermore, based on the new problem definition, we propose a gradient-based **M**ulti-objective optimization **I**nformation **B**ottleneck algorithm, called **MIB**, which introduces a gradient mapping mechanism to ensure that the model can effectively balance the weight allocation and optimization among multiple objectives along the non-convex Pareto front. Furthermore, we conduct several experiments to compare MIB with existing IB methods and show that MIB achieves state-of-the-art performance.

Our contributions can be summarized as follows:

- *New problem and insight*: we deeply analyze the potential drawbacks of existing information bottleneck optimization methods, and experimentally show the complexity and inaccuracy of manually finding Lagrange multipliers.

- *New optimization objective*: for the first time, we examine the information bottleneck problem from the multi-objective perspective, and redefine the trade-off between compression and prediction objectives as a rigorous multi-objective optimization problem.

- *New learning paradigm*: we propose a gradient-based **M**ulti-objective optimization **IB** learning paradigm (MIB), which ensures that we can find the trade-off between multiple objectives in the optimal optimization direction.

- *Compelling empirical results*: extensive experiments demonstrate that our method achieves higher performance ceilings and finds a more stable and optimal trade-off compared to manual-based information bottleneck methods.

# 2 RETHINKING LAGRANGE MULTIPLIERS IN INFORMATION BOTTLENECK: IS MANUAL TRADE-OFF TRULY OPTIMAL?

Previous research on the information bottleneck (IB) optimization problem has typically adhered to a common paradigm: *transforming the IB problem into its Lagrangian relaxation form using the method of Lagrange multipliers, and empirically adjusting the trade-off between compression and prediction through multiple experimental iterations*. While finding the appropriate Lagrange multiplier is undoubtedly exciting, it also prompts us to question whether this empirical manual trade-off is truly the optimal choice. In this section, we will further analyze the existing paradigm. Figure 2 illustrates the challenges associated with Lagrange multipliers in the Information Bottleneck, where the regulation of Lagrange multipliers in neural networks dramatically affects the mutual information between representations, objectives, and data, significantly impacting model performance and generalization ability, as demonstrated in numerous applications. Inaccurate $\beta$ values may compromise the model's trade-off between compression and prediction.

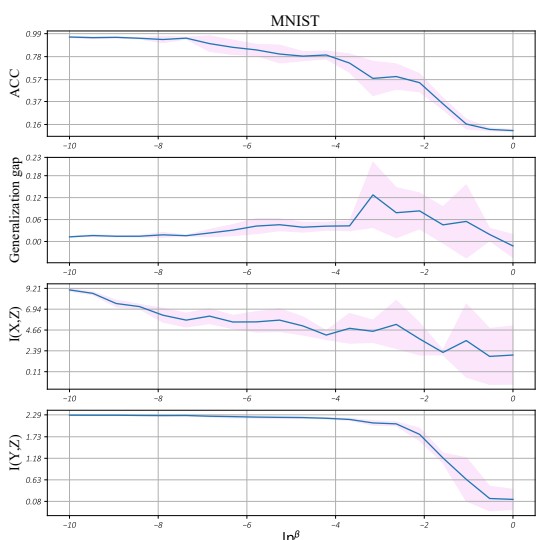

Figure 2: Different $\beta$ values influence the trade-off between compression, prediction, and robustness.

## 2.1 PROBLEM REFORMULATION

To comprehensively understand the trade-off between compression and prediction, we reformulate the Information Bottleneck problem as a multi-objective optimization problem. Here, we recall the classic Information Bottleneck objective function:

$$\min_{p(Z|X)} I(X;Z) - \beta I(Y;Z), \tag{1}$$

where $I(X;Z)$ represents the mutual information between input $X$ and latent representation $Z$, measuring compression loss; $I(Y;Z)$ represents the mutual information between target $Y$ and latent representation $Z$, measuring predictive power. $\beta$ is the trade-off parameter balancing these two terms.

Now, we reformulate Equation 1 as a multi-objective optimization problem. Given input variable $X \in \mathcal{X}$, target variable $Y \in \mathcal{Y}$, and latent representation $Z \in \mathcal{Z}$, our goal is to find a set of Pareto-optimal solutions in the probability distribution space $\mathcal{P}(\mathcal{Z}|\mathcal{X})$ that simultaneously minimize the following two objective functions:

$$\min_{p(Z|X) \in \mathcal{P}(\mathcal{Z}|\mathcal{X})} \mathbf{L}(Z) = (I(X;Z), -I(Y;Z)) \tag{2}$$

Here, $I(X;Z)$ corresponds to compression loss, i.e., the information loss from input $X$ to latent representation $Z$; $-I(Y;Z)$ corresponds to prediction loss, i.e., the negative mutual information from latent representation $Z$ to target $Y$. We multiply the second term by -1 to transform it into a minimization problem. Equation 2 explicitly describes the multi-objective nature of the IB problem,

namely minimizing mutual information between input and latent representation while maximizing mutual information between latent representation and target.

Unlike the original single-objective formulation (Equation 1), the multi-objective formulation (Equation 2) treats compression loss and prediction loss as two independent optimization objectives, rather than combining them into a single scalar through the trade-off parameter $\beta$. This allows us to directly search for and approximate the complete Pareto frontier describing the optimal compression-prediction trade-off, rather than obtaining only a single trade-off point corresponding to a specific $\beta$ value. In the next section, we will formally define the Pareto-optimal frontier for the Information Bottleneck and discuss the limitations of the traditional Lagrangian relaxation method.

## 2.2 PARETO-OPTIMAL FRONTIER

Based on the above multi-objective optimization problem, we can formally define the Pareto-optimal frontier for the Information Bottleneck. Let $\mathcal{F} \subseteq \mathbb{R}^2$ represent the objective space, where each point $\mathbf{l} = (I(X;Z), -I(Y;Z)) \in \mathcal{F}$ represents the loss values of a feasible solution on the two objectives. We say that point $\mathbf{l}^1$ dominates point $\mathbf{l}^2$ if and only if $I^1(X;Z) \leq I^2(X;Z)$ and $I^1(Y;Z) \geq I^2(Y;Z)$, with at least one inequality being strict. A point $\mathbf{l}^* \in \mathcal{F}$ is called Pareto-optimal if it is not dominated by any other point. The Pareto-optimal frontier $\mathcal{F}^*$ for the information bottleneck is the set of all Pareto-optimal points:

$$\mathcal{F}^* = \{\mathbf{l}^* \in \mathcal{F} | \mathbf{l} \in \mathcal{F}, \text{ such that } \mathbf{l} \text{ dominates } \mathbf{l}^*\}. \tag{3}$$

The Pareto frontier delineates the optimal trade-off curve between compression and prediction losses. It provides a global perspective for analyzing the IB problem, revealing the inherent conflicts and dependencies between the two objectives. Moreover, each solution on the frontier corresponds to specific task requirements or preferences, thus having practical significance in different application scenarios. This aligns with the motivation of the Information Bottleneck, which is to maximize predictive performance under given compression constraints, or to minimize compression loss under given predictive performance requirements.

## 2.3 LIMITATIONS OF THE TRADITIONAL APPROACH

Here, we revisit the **challenges** faced by the Lagrangian relaxation method in finding the Pareto frontier:

- *Discrete Trade-offs*. The Lagrangian relaxation simplifies the original problem to a single-objective optimization but can only produce discrete solutions corresponding to specific $\beta$ values. As $\beta$ varies in the range $[0, +\infty)$, the generated solutions constitute a discrete subset of the Pareto frontier rather than a continuous curve. This may lead to suboptimal trade-off choices, especially when the best trade-off lies between discrete solutions.

- *Approximation Incompleteness*. Even with an exhaustive search over $\beta$, the set of solutions $\mathcal{S}_\beta = \{\mathbf{l}_\beta | \beta \in [0, +\infty)\}$ generated by the Lagrangian relaxation method may not well approximate the true Pareto frontier $\mathcal{F}^*$. On the one hand, for many non-convex practical problems, the Lagrangian relaxation method may fail to find Pareto-optimal solutions located in concave regions. On the other hand, the mapping from $\beta$ values to their induced trade-off points is often highly nonlinear and irreversible, making it difficult to control the distribution and coverage of trade-off points by adjusting $\beta$.

To overcome these limitations, we propose a new multi-objective optimization method aimed at directly searching for and generating optimal solutions on the Pareto frontier without relying on preset trade-off parameters. This method can adapt to complex frontier shapes, providing decision-makers with comprehensive and fine-grained trade-off choices.

## 3 MIB FOR FINDING THE PARETO OPTIMAL SOLUTION

In this section, we will detail our proposed Multi-objective Information Bottleneck (MIB) method. MIB aims to address the limitations of manually adjusting Lagrange multipliers in traditional information bottleneck methods by automatically finding the optimal balance between compression and prediction objectives through multi-objective optimization techniques.

To apply the multi-objective optimization problem (Equation 2) in practice, we need to transform the mutual information objectives into optimizable loss functions. Under the variational inference framework, we introduce a parameterized encoder $p_\phi(Z|X)$ to approximate the true posterior distribution $p(Z|X)$, and a parameterized decoder $q_\theta(Y|Z)$ to approximate the true conditional distribution $p(Y|Z)$. Here, $\phi$ and $\theta$ represent the learnable parameters of the encoder and decoder, respectively. Using these variational approximations, we can rewrite the mutual information objectives as:

$$
\begin{aligned}
I(X;Z) &= \mathbb{E}_{p(X)}[D_{\mathrm{KL}}(p(Z|X) \parallel q(Z))] \\
&\approx \mathbb{E}_{p(X)}[D_{\mathrm{KL}}(p_\phi(Z|X) \parallel q(Z))],
\end{aligned}
\tag{4}
$$

$$
\begin{aligned}
I(Y;Z) &= H(Y) - H(Y|Z) \\
&= H(Y) + \mathbb{E}_{p(Y,Z)}[\log p(Y|Z)] \\
&\approx H(Y) + \mathbb{E}_{p(X,Y)}[\mathbb{E}_{p_\phi(Z|X)}[\log q_\theta(Y|Z)]].
\end{aligned}
\tag{5}
$$

In Equation 4, we approximate the true posterior distribution $p(Z|X)$ with the parameterized encoder $p_\phi(Z|X)$ and set the prior distribution $q(Z)$ to be a standard normal distribution $\mathcal{N}(\mathbf{0}, \mathbf{I})$. In Equation 5, we approximate the true conditional distribution $p(Y|Z)$ with the parameterized decoder $q_\theta(Y|Z)$ and ignore the constant term $H(Y)$ which is irrelevant to optimization. Substituting Equations 4 and 5 into the multi-objective optimization problem (Equation 2), we obtain the variational objective for MIB:

$$
\begin{aligned}
\min_\phi \quad & \mathbf{L}(Z) = (\mathcal{L}_{\mathrm{info}}(\phi), \mathcal{L}_{\mathrm{pred}}(\phi, \theta)) \\
\text{where} \quad & \mathcal{L}_{\mathrm{info}}(\phi) = \mathbb{E}_{p(X)}[D_{\mathrm{KL}}(p_\phi(Z|X) \parallel \mathcal{N}(\mathbf{0}, \mathbf{I}))], \\
& \mathcal{L}_{\mathrm{pred}}(\phi, \theta) = -\mathbb{E}_{p(X,Y)}[\mathbb{E}_{p_\phi(Z|X)}[\log q_\theta(Y|Z)]].
\end{aligned}
\tag{6}
$$

Based on the variational objective (Equation 6), we can optimize the parameters of the encoder and decoder using gradient descent algorithms. For each objective function $\mathcal{L}_i(Z)$, we calculate its corresponding gradient:

$$
\mathbf{g}_i = \nabla_\phi \mathcal{L}_i(\phi), \quad i \in \{\mathrm{info}, \mathrm{pred}\}.
\tag{7}
$$

We use a variant of the Frank-Wolfe algorithm Frank & Wolfe (1956) to find the minimum norm element of the gradient vectors. This process can be formalized as the following optimization problem:

$$
\begin{aligned}
\mathbf{w}^* &= \arg\min_{\mathbf{w}} \| \sum_{i \in \{\mathrm{pred}, \mathrm{info}\}} w_i \mathbf{g}_i \|^2, \\
\text{s.t.} & \sum_{i \in \{\mathrm{pred}, \mathrm{info}\}} w_i = 1, w_i \geq 0
\end{aligned}
\tag{8}
$$

The Frank-Wolfe algorithm approximates the solution to the above optimization problem by iteratively solving linear programming problems. At the $t$-th iteration:

(i) Compute the gradient:

$$
\mathbf{d}_t = \nabla_{\mathbf{w}} \| \sum_i w_i \mathbf{g}_i \|^2 |_{\mathbf{w} = \mathbf{w}_t}
\tag{9}
$$

(ii)) Solve the linear programming problem:

$$
\mathbf{s}_t = \arg\min_{\mathbf{s}} \{ \mathbf{d}_t^T \mathbf{s} : \sum_i s_i = 1, s_i \geq 0 \}
\tag{10}
$$

(iii) Update the weights:

$$
\mathbf{w}_{t+1} = (1 - \gamma_t)\mathbf{w}_t + \gamma_t \mathbf{s}_t, \quad \text{where} \gamma_t = \frac{2}{t+2}
\tag{11}
$$

The iteration continues until convergence or reaching the maximum number of iterations. The final $\mathbf{w}^*$ gives the optimal weight combination for the objective functions. To enhance the flexibility of optimization, we introduce a nonlinear weight combination:

$$
\tilde{w}_i = (w_i^*)^{1 + \sqrt{\mathrm{sqrted}}}, \quad i \in \{\mathrm{pred}, \mathrm{info}\}
\tag{12}
$$

where sqrted is a boolean value controlling whether to apply the nonlinear transformation. This nonlinear transformation allows for more flexible adjustment of the trade-off between different objectives.

To improve the stability of optimization, we adopt the Smooth Information Bottleneck (Smooth-IB) method. Given the weighted loss functions $\tilde{\mathcal{L}}_i = \tilde{w}_i \mathcal{L}_i(Z)$, the Smooth-IB method defines a new loss function:

$$\mathcal{L}_{\text{Smooth-IB}} = \sum_{i \in \{\text{pred},\text{info}\}} \tilde{\mathcal{L}}_i + \mu \log \left( \sum_{i \in \{\text{pred},\text{info}\}} \exp \left( \frac{\tilde{\mathcal{L}}_i - \min_j \tilde{\mathcal{L}}_j}{\mu} \right) \right) + \min_j \tilde{\mathcal{L}}_j \qquad (13)$$

where $\mu > 0$ is a temperature parameter controlling the smoothness of the loss function, typically fixed at 0.2. The gradient of the Smooth-IB loss can be expressed as:

$$\nabla \mathcal{L}_{\text{Smooth-IB}} = \sum_{i \in \{\text{pred},\text{info}\}} \tilde{w}_i \nabla \mathcal{L}_i(Z) + \sum_{i \in \{\text{pred},\text{info}\}} \frac{\exp \left( \frac{\tilde{\mathcal{L}}_i - \min_j \tilde{\mathcal{L}}_j}{\mu} \right)}{\sum_{k \in \{\text{pred},\text{info}\}} \exp \left( \frac{\tilde{\mathcal{L}}_k - \min_j \tilde{\mathcal{L}}_j}{\mu} \right)} \tilde{w}_i \nabla \mathcal{L}_i(Z)$$
$$(14)$$

This gradient form ensures that all objectives receive appropriate attention during the optimization process while providing a smoother optimization surface. We propose a proposition to summarize that our method can effectively find the Pareto front in Equation 6. For the proof, please refer to **Appendix A**.

**Proposition 1** (Pareto Optimality). *Let $\mathcal{F}^*$ be the true Pareto front, defined as:*

$$\mathcal{F}^* = \{(\mathcal{L}_{info}(\phi), \mathcal{L}_{pred}(\phi,\theta)) | \phi', \theta' : \mathcal{L}_{info}(\phi') \le \mathcal{L}_{info}(\phi) \wedge \mathcal{L}_{pred}(\phi',\theta') \le \mathcal{L}_{pred}(\phi,\theta)\}$$

*Let $\mathcal{S}_{MIB}$ be the set of solutions generated by the MIB method. Then for any $\varepsilon > 0$, there exists a sufficiently large number of iterations $T$, such that:*

$$\forall s \in \mathcal{S}_{MIB}, \exists s^* \in \mathcal{F}^* : \|s - s^*\|_2 < \varepsilon$$

*where $\| \cdot \|_2$ denotes the Euclidean norm.*

## 4 EXPERIMENTS

In this section, we conduct empirical evaluations on mainstream datasets to demonstrate the superiority of the proposed MIB.

### 4.1 EXPERIMENT SETTINGS

**Datasets.** We applied our method to FashionMNIST Xiao et al. (2017) and CIFAR-10 Krizhevsky et al. (2009), achieving consistent Pareto-optimal parameter adaptation across both datasets. FashionMNIST is a 10-category grayscale image dataset, and CIFAR-10 is a diverse 10-class color image dataset with various objects, which are crucial for benchmarking image classification and analyzing performance trends.

**Baselines and Adversarial Attacks.** We select a variety of IB methods as baselines, which are based on different theoretical ideas, including RES18, VIB, NIB, VIB-squared, and NIB-squared. For more detailed descriptions of these methods, please refer to **Appendix D**. In addition, to better measure the performance and robustness of different IB methods, we introduce a variety of attack methods. Depending on their complexity, these attacks can be classified as simple attacks, e.g., [FGSM] Goodfellow et al. (2014), [PGD] Mądry et al. (2017), and complex attacks, e.g., [NIFGSM] Lin et al. (2019), [EOTPGD] Liu et al. (2018), [MIFGSM] Dong et al. (2018), [UPGD] Mądry et al. (2017), [Jitter] Schwinn et al. (2023). Here, we give a brief explanation of these attacks:

- FGSM (Fast Gradient Sign Method): This is a quick method that perturbs images by adding noise in the direction of the gradient of the loss with respect to the input image.

- PGD (Projected Gradient Descent): A more sophisticated method than FGSM, it iteratively applies small perturbations and projects the perturbed image back to a valid range after each step.

- NIFGSM (Nesterov Iterative Fast Gradient Sign Method): An iterative method similar to FGSM but includes Nesterov accelerated gradient, which can enhance the effectiveness of the attack.

- EOTPGD (Expectation Over Transformation Projected Gradient Descent): A variant of PGD that considers transformations like rotation or translation of images during the attack process.

- MIFGSM (Momentum Iterative Fast Gradient Sign Method): Integrates momentum into the iterative FGSM process to stabilize update directions and improve the attack's success rate.

- UPGD (Universal Perturbation Gradient Descent): Aims to find universal perturbations applicable to a wide range of inputs.

- Jitter: Involves adding random noise (jitter) to images to test the model's sensitivity to small variations.

**Implementation.** We evaluate the effectiveness of our proposed multi-objective optimization method for information bottleneck by conducting comparisons with several established Lagrangian variants of IB. Optimal settings for the Lagrange multiplier method, as sourced from prior studies, have been adopted. Our approach automatically uncovers the Pareto-optimal balance between predictive accuracy and data compression, eliminating the need for manual tuning of $\beta$. Model performance is gauged in terms of both generalization and robustness to adversarial attacks. We employ ResNet-18 as the backbone model, with a three-layer MLP to facilitate the computation of mutual information and to perform classification. Details concerning the model and experimental procedures are furnished in **Appendix C**.

## 4.2 BENCHMARK RESULTS

**Results on FashionMNIST.** The empirical analysis conducted on the FashionMNIST dataset, as summarized in Table 1, provides a comprehensive evaluation of our method against a suite of adversarial attacks. The performance metrics reveal the superior robustness and efficacy of MIB in comparison to other approaches that manually fix $\beta$.

In the non-adversarial setting [NONE], the MIB model exhibits a marginally higher accuracy than the mainstream VIB model. Nevertheless, this minor trade-off is substantially outweighed by the significant gains in robustness across all adversarial scenarios tested. Notably, under [FGSM], MIB shows an improvement of 4.06% over the next best method (NIB), establishing its proficiency in countering this attack. The advantage of MIB becomes more pronounced under sophisticated iterative attack schemes. For example, against [PGD], MIB maintains a 4.57% higher accuracy than VIB-squared. Under the [EOTPGD] and [UPGD] attacks, MIB demonstrates remarkable resilience, proving its enhanced robust feature extraction capability. The most compelling evidence of MIB's robustness is its performance under the MIFGSM and Jitter attacks. These results underscore the effectiveness of the MIB method in navigating the complex trade-off between accuracy and robustness.

**Results on CIFAR-10.** We also conduct experiments on the CIFAR-10 dataset to determine the advantages of our proposed MIB method against various adversarial attacks, and the results are shown in Table 2. It can be found that in the absence of adversarial attacks [NONE], the MIB model achieves a performance close to SOTA in accuracy. When facing more lenient attacks, e.g., [FGSM] and [PGD], MIB significantly outperforms other IB methods, especially outperforming the standard VIB model by about 10% under [PGD].

Furthermore, by analyzing the sophisticated [NIFGSM], [EOTPGD], [MIFGSM], [UPGD], and [Jitter] attacks, we can gain more insight into the resilience of MIB. The experimental results also show that MIB performs as well as ever in the face of both simple and complex attacks, which indicates that MIB can effectively resist strong attacks such as [Jitter] attack and demonstrate the robustness of MIB. More importantly, the standard deviation of MIB is lower than that of NIB-squared, which is the closest method to MIB, reflecting the stability of MIB under random conditions.

These experimental results, when holistically reviewed, not only validate the superior robustness of MIB against a wide spectrum of adversarial attacks but also showcase its ability to maintain high accuracy with less variability in performance. This balance between precision and stability exemplifies the model's capacity to handle adversarial perturbations.

Table 1: Comparative analysis of adversarial defense methods on the FashionMNIST dataset. **Blod** indicates the best performance while underline indicates the second best. We report the average results of three random trials.

| Attack | RES18 | VIB | NIB | VIB-squared | NIB-squared | MIB |
|---|---|---|---|---|---|---|
| NONE | 87.20 ±0.25 | 87.43 ±0.13 | 86.80 ±0.14 | 84.58 ±1.10 | 87.24 ±0.40 | **87.67 ±0.08** |
| FGSM Goodfellow et al. (2014) | 49.23 ±1.83 | 52.33 ±1.57 | 53.01 ±0.90 | 52.09 ±1.07 | 52.85 ±1.21 | **57.07 ±2.01** |
| PGD Mądry et al. (2017) | 39.37 ±2.14 | 42.53 ±2.06 | 37.94 ±1.89 | 46.70 ±1.84 | 42.37 ±1.16 | **51.27 ±2.53** |
| NIFGSM Lin et al. (2019) | 60.28 ±1.04 | 62.48 ±1.31 | 63.65 ±1.22 | 63.32 ±1.50 | 65.53 ±1.14 | **67.07 ±1.04** |
| EOTPGD Liu et al. (2018) | 38.42 ±2.14 | 41.84 ±2.26 | 37.81 ±1.63 | 46.75 ±2.00 | 42.49 ±1.10 | **50.73 ±2.62** |
| MIFGSM Dong et al. (2018) | 24.43 ±2.87 | 33.40 ±4.02 | 32.16 ±2.05 | 40.69 ±3.10 | 42.97 ±0.96 | **45.38 ±3.56** |
| UPGD Mądry et al. (2017) | 21.60 ±2.82 | 30.94 ±4.23 | 27.49 ±2.27 | 39.44 ±3.11 | 38.02 ±1.15 | **44.36 ±1.61** |
| Jitter Schwinn et al. (2023) | 41.99 ±1.90 | 45.38 ±1.86 | 44.54 ±1.44 | 48.07 ±1.42 | 50.34 ±1.02 | **53.76 ±2.16** |
| Rank Avg. | 5.50 | 3.75 | 4.63 | 3.50 | 2.63 | 1.00 |

Table 2: Comparative analysis of adversarial defense methods on the CIFAR-10 dataset. **Blod** indicates the best performance while underline indicates the second best. We report the average results of three random trials.

| Attack | RES18 | VIB | NIB | VIB-squared | NIB-squared | MIB |
|---|---|---|---|---|---|---|
| NONE | 66.32 ±0.55 | 66.59 ±1.48 | 66.91 ±0.35 | 62.38 ±0.23 | **67.38 ±1.62** | 66.96 ±0.79 |
| FGSM Goodfellow et al. (2014) | 44.06 ±0.54 | 44.14 ±1.14 | 44.89 ±0.57 | 40.84 ±0.21 | 45.68 ±1.23 | **46.28 ±0.52** |
| PGD Mądry et al. (2017) | 42.57 ±0.57 | 42.70 ±1.11 | 43.25 ±0.37 | 39.61 ±0.39 | 44.13 ±1.38 | **44.93 ±0.54** |
| NIFGSM Lin et al. (2019) | 52.69 ±0.51 | 52.86 ±1.32 | 54.05 ±0.37 | 49.49 ±0.09 | 54.37 ±1.60 | **55.06 ±0.55** |
| EOTPGD Liu et al. (2018) | 42.50 ±0.67 | 42.63 ±1.16 | 43.34 ±0.46 | 39.60 ±0.39 | 43.90 ±1.11 | **44.70 ±0.46** |
| MIFGSM Dong et al. (2018) | 40.62 ±0.79 | 40.82 ±1.14 | 41.53 ±0.67 | 37.54 ±0.19 | 42.61 ±1.21 | **43.47 ±0.74** |
| UPGD Mądry et al. (2017) | 40.63 ±0.86 | 40.85 ±1.17 | 41.74 ±0.61 | 37.91 ±0.43 | 42.41 ±1.13 | **43.57 ±0.66** |
| Jitter Schwinn et al. (2023) | 42.65 ±0.61 | 42.79 ±1.06 | 43.80 ±0.50 | 39.43 ±0.17 | 44.63 ±1.31 | **45.93 ±0.62** |
| Rank Avg. | 5.00 | 4.00 | 3.00 | 6.00 | 1.88 | 1.13 |

### 4.3 FURTHER ANALYSIS

In this section, we conduct a detailed analysis of the mechanism of MIB and discuss the following issues. All the analysis experiments are conducted on FashionMNIST. More empirical results are reported in **Appendix E**.

> **Q1:** Why is MIB a better choice for finding complex trade-offs than other IB methods?

Figure 3 illustrates the advantages of our MIB over linear weighting in searching for the Pareto front of the Information Bottleneck. From left to right, the figures correspond to Pareto fronts of different shapes, all of which are represented in Figure 1. When the shape of the Pareto front is convex, corresponding to Figure 3(a) and the first subplot in 1, linear weighting can effectively search the Pareto front (although this may *rely on experience or extensive manual experimentation*). However, when the Pareto front has complex shapes or non-convex regions, as shown in Figure 3(b), linear weighting fails to find certain solutions on the Pareto front because this method can only search for points tangent to the front (blue dotted line). In fact, such complex scenarios are likely to occur in the broad context of deep learning (as represented by the third subplot in Figure 1). Our method MIB, through gradient projection and reconciliation, allows for a freer descent direction than linear weighting. The joint descent direction for both objectives enables us to find solutions on complex-shaped Pareto fronts, as shown in Figure 3(c). In **Appendix A**, we provide detailed proof that our method can converge to the Pareto front and search regions that linear weighting cannot reach.

> **Q2:** Does MIB effectively find the complex trade-off between compression and prediction?

The t-SNE visualization in Figure 4 reflects the multi-objective optimization strengths of our method within the Information Bottleneck (IB) framework, compared to the traditional manual-based IB

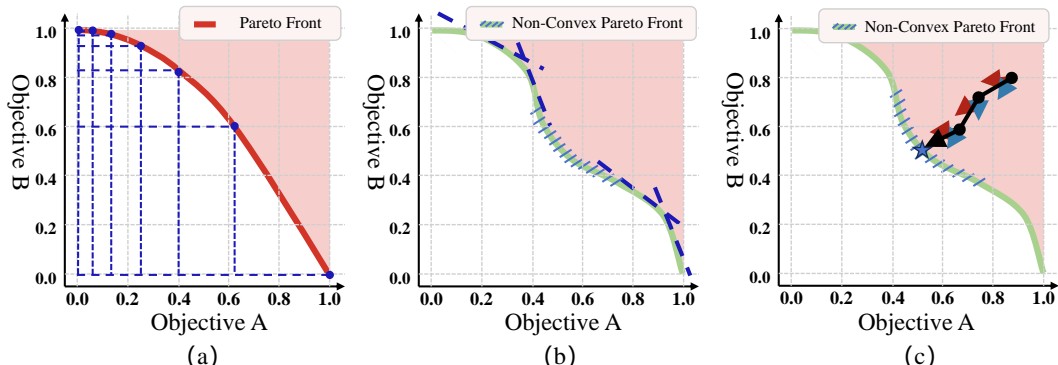

Figure 3: The difference between linear weighting-based IB and multi-objective optimization IB on Pareto fronts of different shapes. (a) Effectiveness of conventional IB on the convex Pareto front. (b) Limitations of the conventional IB on the non-convex Pareto front. (c) MIB can find solutions on complex-shaped Pareto fronts.

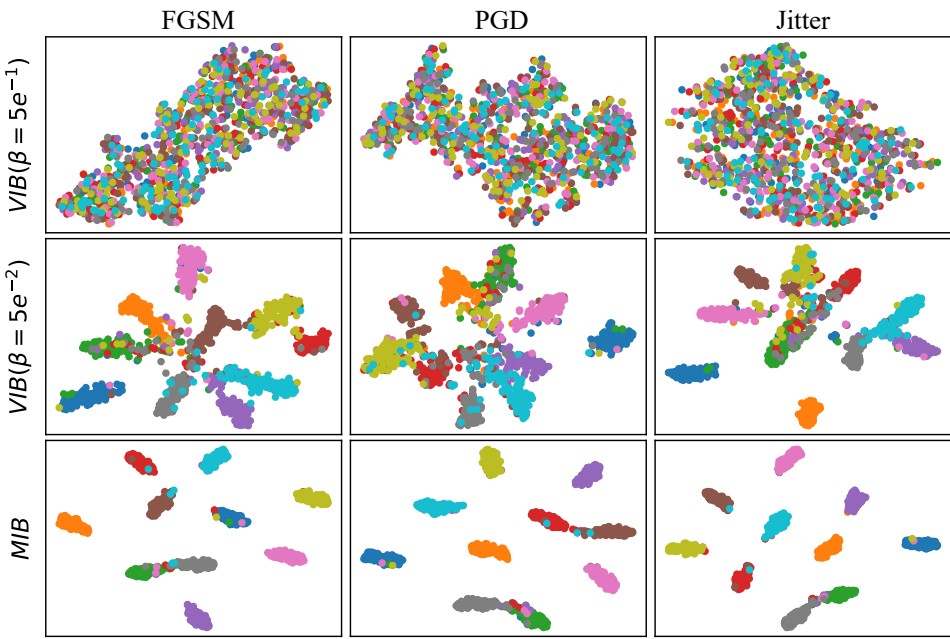

Figure 4: t-SNE visualization of MIB and VIB (with two $\beta$ settings) under different adversarial attacks [FGSM], [PGD], and [Jitter].

methods. MIB, our method, achieves distinctly separated data clusters, even under adversarial attack conditions like [FGSM], [PGD], and [Jitter], which suggests an effective trade-off between information compression and prediction. This separation exemplifies how our multi-objective approach not only maintains the integrity of the latent space but also ensures robustness, a key advantage over traditional IB methods that rely on a fixed beta for the trade-off. Our method's adaptability to automatically tune this balance is a significant leap toward practical and resilient machine learning models.

> **Q3:** What are the key factors that make MIB efficient?

Figure 5 compares the robustness of different Information Bottleneck (IB) methods against various adversarial attacks. It demonstrates that the Multi-objective Information Bottleneck (MIB) method,

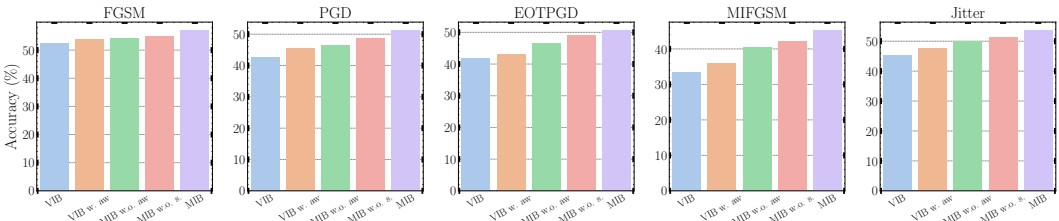

Figure 5: Ablation experiment results. The methods compared include Variational Information Bottleneck (VIB), VIB with automatic weight adjustment (VIB w. aw), Multi-objective IB without automatic weight adjustment (MIB w.o. aw), MIB without smoothing (MIB w.o. s.), and the full MIB method.

which incorporates automatic weight adjustment, multi-objective optimization, and smoothing techniques, progressively approaches the Pareto frontier. The complete MIB method shows the best performance across most attack types, highlighting its effectiveness in balancing compression and prediction objectives and achieving superior robustness against adversarial attacks.

## 5 CHALLENGES AND FUTURE

The Information Bottleneck (IB) provides a profound perspective for deep learning. On the one hand, incorporating the IB principle has led to significant improvements across numerous domains. More importantly, IB offers a multitude of insights and considerations for deciphering the "black box" of neural networks. Nevertheless, the IB approach still faces several limitations and challenges, such as accurately estimating mutual information, delving deeper into the profound implications of IB, and providing more comprehensive explanations from an optimization standpoint. We attempt to re-evaluate the trade-off inherent in the IB from a multi-objective perspective. However, to pursue deeper exploration in this direction, a more precise understanding of this series of issues is requisite. In subsequent work, we will undertake a more comprehensive investigation of these matters.

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

# Appendix
# Exploring Complex Trade-offs in Information Bottleneck through Multi-Objective Optimization

The content of the **Appendix** is summarized as follows:

1) in Sec. A, we provide rigorous proofs to support the relevant definitions in the main paper.

2) in Sec. B, we briefly present the state of the art in the field of information bottlenecks.

3) in Sec. C, we make further additions to the aforementioned experimental details.

4) in Sec. D, we demonstrate the details of baselines we use in experiments of MIB.

5) in Sec. E, we illustrate more detailed empirical results and analyses of MIB.

6) in Sec. **??**, we present the related work on multi-objective optimization.

7) in Sec. G, to better illustrate the execution process of MIB, we provide a detailed execution flow in Algorithm 1.

## A    DETAILED PROOF

**Proposition 2** (Pareto Optimality). *Let $\mathcal{F}^*$ be the true Pareto frontier, defined as:*

$$\mathcal{F}^* = \{(\mathcal{L}_{info}(\phi), \mathcal{L}_{pred}(\phi, \theta)) | \phi', \theta' : \mathcal{L}_{info}(\phi') \leq \mathcal{L}_{info}(\phi) \wedge \mathcal{L}_{pred}(\phi', \theta') \leq \mathcal{L}_{pred}(\phi, \theta)\}$$

*Let $\mathcal{S}_{MIB}$ be the set of solutions generated by the MIB method. Then for any $\varepsilon > 0$, there exists a sufficiently large number of iterations $T$ such that:*

$$\forall s \in \mathcal{S}_{MIB}, \exists s^* \in \mathcal{F}^* : \|s - s^*\|_2 < \varepsilon$$

*where $\| \cdot \|_2$ denotes the Euclidean norm.*

**Theorem 1** (Pareto Optimality and Invariance of the MIB Method). *Let $\mathcal{F}^*$ be the true Pareto front, $\mathcal{S}_{MIB}$ be the solution set generated by the MIB method, $\mathbf{w}^*$ be the optimal weight vector obtained by the Frank-Wolfe algorithm, and $\tilde{\mathbf{w}}$ the weight vector after a nonlinear transformation. Then:*

- ***Pareto Optimality**: $\forall \varepsilon > 0, \exists T \in \mathbb{N}, \forall t > T : \forall s_t \in \mathcal{S}_{MIB}, \exists s^* \in \mathcal{F}^* : \|s_t - s^*\|_2 < \varepsilon$*

- ***Invariance**: $\forall \mu \geq 0, \mathcal{P}(\mathcal{L}_{MIB}(\mathbf{w}^*)) = \mathcal{P}(\mathcal{L}_{Smooth\text{-}IB}(\tilde{\mathbf{w}}))$*

*where $\mathcal{P}(\cdot)$ denotes the Pareto optimal solution set of an optimization problem.*

**Proof.  Part 1: Pareto Optimality**

1. Define the true Pareto front $\mathcal{F}^*$:

$$\begin{aligned}
\mathcal{F}^* = \big\{(\mathcal{L}_{\text{info}}(\phi), \mathcal{L}_{\text{pred}}(\phi, \theta)) \in \mathbb{R}^2 \mid (\phi', \theta') : \\
\mathcal{L}_{\text{info}}(\phi') \leq \mathcal{L}_{\text{info}}(\phi) \wedge \mathcal{L}_{\text{pred}}(\phi', \theta') \leq \mathcal{L}_{\text{pred}}(\phi, \theta) \wedge \\
(\mathcal{L}_{\text{info}}(\phi'), \mathcal{L}_{\text{pred}}(\phi', \theta')) \neq (\mathcal{L}_{\text{info}}(\phi), \mathcal{L}_{\text{pred}}(\phi, \theta))\big\}
\end{aligned} \tag{15}$$

2. Frank-Wolfe algorithm iterative process: at the $k$-th iteration, the weight vector is updated as:

$$\mathbf{w}^{(k+1)} = (1 - \gamma_k)\mathbf{w}^{(k)} + \gamma_k \mathbf{s}^{(k)} \tag{16}$$

where $\gamma_k = \frac{2}{k+2}$, and $\mathbf{s}^{(k)} = \arg\min_{\mathbf{s} \in \Delta^2} \langle \mathbf{s}, \nabla f(\mathbf{w}^{(k)}) \rangle$.

Convergence:

$$\|\mathbf{w}^{(k)} - \mathbf{w}^*\|_2 \leq \frac{2D}{k+2} \tag{17}$$

where $D$ is the problem's diameter.

3. Define $\delta$-approximate Pareto optimal solution:

$$
\begin{aligned}
\mathcal{F}_\delta^* = \{(&\mathcal{L}_{\text{info}}(\phi), \mathcal{L}_{\text{pred}}(\phi, \theta)) \mid (\phi', \theta') : \\
&\mathcal{L}_{\text{info}}(\phi') \leq \mathcal{L}_{\text{info}}(\phi) - \delta \wedge \mathcal{L}_{\text{pred}}(\phi', \theta') \leq \mathcal{L}_{\text{pred}}(\phi, \theta) - \delta \\
&\}
\end{aligned}
\tag{18}
$$

4. MIB method iterative process: at the $t$-th iteration, the parameters $(\phi_t, \theta_t)$ are updated as:

$$
(\phi_{t+1}, \theta_{t+1}) = (\phi_t, \theta_t) - \eta_t \nabla_{\phi, \theta} \mathcal{L}_{\text{Smooth-IB}}(\tilde{\mathbf{w}}, \phi_t, \theta_t)
\tag{19}
$$

where $\eta_t$ is the learning rate.

5. **Lemma**: $\forall \delta > 0, \exists T_\delta \in \mathbb{N}, \forall t > T_\delta : s_t \in \mathcal{F}_\delta^*$

   *Proof of Lemma*: By contradiction. Suppose $\exists \delta_0 > 0, \forall T \in \mathbb{N}, \exists t > T : s_t \notin \mathcal{F}_{\delta_0}^*$.

   This implies that $\exists(\phi', \theta')$, such that:

$$
\mathcal{L}_{\text{info}}(\phi') \leq \mathcal{L}_{\text{info}}(\phi_t) - \delta_0 \quad \text{and} \quad \mathcal{L}_{\text{pred}}(\phi', \theta') \leq \mathcal{L}_{\text{pred}}(\phi_t, \theta_t) - \delta_0
\tag{20}
$$

   By the definition of $\mathcal{L}_{\text{Smooth-IB}}$, we have:

$$
\mathcal{L}_{\text{Smooth-IB}}(\tilde{\mathbf{w}}, \phi', \theta') < \mathcal{L}_{\text{Smooth-IB}}(\tilde{\mathbf{w}}, \phi_t, \theta_t) - \min(\tilde{w}_{\text{info}}, \tilde{w}_{\text{pred}})\delta_0
\tag{21}
$$

   This contradicts the convergence properties of the MIB method. Thus, the assumption is false.

6. From the lemma and the definition of $\mathcal{F}_\delta^*$, we can deduce:

$$
\forall \varepsilon > 0, \exists \delta > 0, T_\delta \in \mathbb{N}, \forall t > T_\delta : \forall s_t \in \mathcal{S}_{\text{MIB}}, \exists s^* \in \mathcal{F}^* : \|s_t - s^*\|_2 < \varepsilon
\tag{22}
$$

**Part 2: Invariance**

1. Define the nonlinear weight transformation:

$$
\tilde{w}_i = g(w_i^*) = (w_i^*)^{1+\sqrt{\text{sqrted}}}, \quad i \in \{\text{pred}, \text{info}\}
\tag{23}
$$

   Monotonicity of the transformation $g$:

$$
\frac{dg}{dw} = (1 + \sqrt{\text{sqrted}})(w^*)^{\sqrt{\text{sqrted}}} > 0, \quad \forall w^* \in (0, 1]
\tag{24}
$$

2. Define $\mathcal{L}_{\text{MIB}}$ and $\mathcal{L}_{\text{Smooth-IB}}$:

$$
\mathcal{L}_{\text{MIB}}(\mathbf{w}^*, \phi, \theta) = w_{\text{pred}}^* \mathcal{L}_{\text{pred}}(\phi, \theta) + w_{\text{info}}^* \mathcal{L}_{\text{info}}(\phi)
\tag{25}
$$

$$
\begin{aligned}
\mathcal{L}_{\text{Smooth-IB}}(\tilde{\mathbf{w}}, \phi, \theta) = &\sum_i \tilde{w}_i \mathcal{L}_i(\phi, \theta) \\
&+ \mu \log\left(\sum_i \exp\left(\frac{\tilde{w}_i \mathcal{L}_i(\phi, \theta) - \min_j(\tilde{w}_j \mathcal{L}_j(\phi, \theta))}{\mu}\right)\right) \\
&+ \min_j(\tilde{w}_j \mathcal{L}_j(\phi, \theta))
\end{aligned}
\tag{26}
$$

3. **Lemma**: $\forall(\phi, \theta), (\phi', \theta')$, if $\mathcal{L}_{\text{MIB}}(\mathbf{w}^*, \phi, \theta) \leq \mathcal{L}_{\text{MIB}}(\mathbf{w}^*, \phi', \theta')$, then $\mathcal{L}_{\text{Smooth-IB}}(\tilde{\mathbf{w}}, \phi, \theta) \leq \mathcal{L}_{\text{Smooth-IB}}(\tilde{\mathbf{w}}, \phi', \theta')$

   *Proof of Lemma*: Use the monotonicity of the nonlinear transformation $g$ and the definition of $\mathcal{L}_{\text{Smooth-IB}}$.

4. Define the Pareto optimal solution set:

$$
\mathcal{P}(\mathcal{L}) = \{(\phi^*, \theta^*) \mid (\phi, \theta) : \mathcal{L}(\phi, \theta) < \mathcal{L}(\phi^*, \theta^*)\}
\tag{27}
$$

5. Assume $(\phi^*, \theta^*) \in \mathcal{P}(\mathcal{L}_{\text{MIB}}(\mathbf{w}^*))$. By contradiction, suppose $(\phi^*, \theta^*) \notin \mathcal{P}(\mathcal{L}_{\text{Smooth-IB}}(\tilde{\mathbf{w}}))$.

Then $\exists(\phi', \theta')$, such that:

$$\mathcal{L}_{\text{Smooth-IB}}(\tilde{\mathbf{w}}, \phi', \theta') < \mathcal{L}_{\text{Smooth-IB}}(\tilde{\mathbf{w}}, \phi^*, \theta^*) \tag{28}$$

According to the lemma, this implies:

$$\mathcal{L}_{\text{MIB}}(\mathbf{w}^*, \phi', \theta') < \mathcal{L}_{\text{MIB}}(\mathbf{w}^*, \phi^*, \theta^*) \tag{29}$$

This contradicts the assumption that $(\phi^*, \theta^*) \in \mathcal{P}(\mathcal{L}_{\text{MIB}}(\mathbf{w}^*))$.

Therefore, $\mathcal{P}(\mathcal{L}_{\text{MIB}}(\mathbf{w}^*)) \subseteq \mathcal{P}(\mathcal{L}_{\text{Smooth-IB}}(\tilde{\mathbf{w}}))$.

6. Similarly, we can prove that $\mathcal{P}(\mathcal{L}_{\text{Smooth-IB}}(\tilde{\mathbf{w}})) \subseteq \mathcal{P}(\mathcal{L}_{\text{MIB}}(\mathbf{w}^*))$.

7. Therefore, $\mathcal{P}(\mathcal{L}_{\text{MIB}}(\mathbf{w}^*)) = \mathcal{P}(\mathcal{L}_{\text{Smooth-IB}}(\tilde{\mathbf{w}}))$.

Thus, the theorem is proven. □

This proof proven that the MIB method not only converges to solutions arbitrarily close to the true Pareto front, but also preserves the Pareto optimal solution set of the original problem through nonlinear weight transformations and the introduction of the SmoothIB loss function. This result emphasizes the MIB method's ability to maintain optimality while improving efficiency and adaptability, especially highlighting its superiority when dealing with non-convex Pareto fronts.

**Proposition 3** (Solutions for Non-Convex Pareto Frontiers). *Let $\mathcal{F}_{nc}^*$ be a non-convex Pareto frontier, $\mathcal{S}_{MNE}$ be the set of solutions obtained using only the minimum norm element method, $\mathcal{S}_{MIB}$ be the set of solutions obtained by the MIB method. Then there exists:*

$$\exists s \in \mathcal{F}_{nc}^* : s \in \mathcal{S}_{MIB} \land s \notin \mathcal{S}_{MNE}$$

*where the MIB method uses nonlinear weight transformation:*

$$\tilde{w}_i = (w_i^*)^{1+\sqrt{sqrted}}, \quad i \in \{pred, info\}$$

*and SmoothIB loss function:*

$$\mathcal{L}_{Smooth-IB} = \sum_i \tilde{\mathcal{L}}_i + \mu \log \left( \sum_i \exp \left( \frac{\tilde{\mathcal{L}}_i - \min_j \tilde{\mathcal{L}}_j}{\mu} \right) \right) + \min_j \tilde{\mathcal{L}}_j$$

**Proof.** We prove this proposition by constructing a counterexample.

1. Consider a non-convex Pareto front $\mathcal{F}_{nc}^*$, which contains a non-convex point $s^* = (x^*, y^*)$.

2. Define the objective function of the Minimum Norm Element (MNE) method:

$$\mathcal{L}_{\text{MNE}}(w, x, y) = w_1 x + w_2 y, \quad \text{where } w_1 + w_2 = 1, w_1, w_2 \geq 0 \tag{30}$$

3. For any weights $w = (w_1, w_2)$, the solution found by the MNE method is always located on the convex hull of the Pareto front. Thus,

$$s^* \notin \mathcal{S}_{\text{MNE}} \tag{31}$$

4. Now consider the MIB method. First, apply a nonlinear weight transformation:

$$\tilde{w}_i = (w_i^*)^{1+\sqrt{sqrted}}, \quad i \in \{1, 2\} \tag{32}$$

5. The MIB method uses the SmoothIB loss function:

$$\mathcal{L}_{\text{Smooth-IB}} = \sum_i \tilde{\mathcal{L}}_i + \mu \log \left( \sum_i \exp \left( \frac{\tilde{\mathcal{L}}_i - \min_j \tilde{\mathcal{L}}_j}{\mu} \right) \right) + \min_j \tilde{\mathcal{L}}_j \tag{33}$$

where $\tilde{\mathcal{L}}_i = \tilde{w}_i \mathcal{L}_i$, $\mathcal{L}_1 = x$, and $\mathcal{L}_2 = y$.

6. There exists a set of weights $w^* = (w_1^*, w_2^*)$ such that at the point $s^*$, $\mathcal{L}_{\text{Smooth-IB}}$ reaches a local minimum. This is because the nonlinearity of the SmoothIB loss function allows it to find a locally optimal solution at a non-convex point.

7. Therefore, there exists a non-convex point $s^*$ such that:

$$s^* \in \mathcal{S}_{\text{MIB}} \tag{34}$$

8. In conclusion, we have found a point $s^* \in \mathcal{F}_{\text{nc}}^*$ that satisfies:

$$s^* \in \mathcal{S}_{\text{MIB}} \wedge s^* \notin \mathcal{S}_{\text{MNE}} \tag{35}$$

Thus, the proposition is proven. $\square$

This proof demonstrates the advantage of the MIB method over the MNE method in handling non-convex Pareto fronts. By using nonlinear weight transformations and the SmoothIB loss function, the MIB method can find Pareto optimal solutions located in non-convex regions, which the MNE method is unable to obtain.

## B    RELATED WORK FOR INFORMATION BOTTLENECK

Recent efforts in Information Bottleneck (IB) theory have addressed optimization challenges, with notable developments including variational approximations Tishby & Zaslavsky (2015), matrix-based entropy functionals Saxe et al. (2019), and invariant input transformations Achille & Soatto (2018). These methods seek to balance compression and prediction accuracy by determining an optimal representation $Z$.

A significant milestone was the Deep Variational Information Bottleneck (VIB) Alemi et al. (2017), which simplifies mutual information estimation using variational techniques and Lagrangian multipliers. This innovation has broadened IB applications in disentangling multi-view data features Bao (2021), fine-tuning language models in low-resource settings Mahabadi et al. (2021), and improving graph neural networks Wu et al. (2020).

Attention has also turned towards the algorithmic determination of parameter $\beta$ to modulate the compression-prediction trade-off. The Elastic Information Bottleneck (EIB) Ni et al. (2022) and Deterministic Information Bottleneck (DIB) Strouse & Schwab (2017) utilize assumptions about data distribution and structure, which can restrict their applicability. Despite the plethora of IB-related research, most studies aim to enhance performance in new domains or through precise information loss functions and mutual information estimation methods. This paper pioneers a multi-objective approach to deeply explore and understand the quintessential trade-off characteristic of IB, marking a novel direction in IB research.

## C    IMPLEMENTATION DETAILS

The experiments were conducted on a server equipped with an NVIDIA Tesla V100S-16GB GPU. The programming language used was Python 3.8, and the primary libraries for deep learning were PyTorch 1.13 with CUDA 11.7 support, and torchattacks 3.5 for implementing adversarial attacks. We utilized a ResNet-18 architecture as the backbone of our models, with a three-layer Multilayer Perceptron (MLP) for the prediction layer. To ensure the robustness and representativeness of the results, the training dataset was evenly split into five distinct subsets, each used both for training and testing the model.

During model training, we employed the Adam optimizer with a learning rate of $1 \times 10^{-3}$, and a batch size of 128. In the Information Bottleneck (IB) method, the bottleneck size $K$ was set to 128, and the $\beta$ parameter was fixed at 0.0001. For the fashion and CIFAR-10 datasets, the number of training epochs was set to 40 and 150, respectively.

Adversarial attacks were parameterized with an intensity parameter $\epsilon$, a step size $\alpha$, and a number of steps; specifically, for the fashion dataset, $\epsilon = \frac{1}{255}, \alpha = \frac{2}{255}$, and steps $= 10$, while for CIFAR-10, $\epsilon = \frac{8}{255}, \alpha = \frac{10}{255}$, and steps $= 10$.

## D    DETAILS OF THE BASELINE

**VIB** Alemi et al. (2017). Variational Information Bottleneck is a method that applies variational inference to optimize the IB Lagrangian that balances the compression and prediction terms. The loss function of VIB is given by:

$$L_{VIB} = \frac{1}{N} \sum_{n=1}^{N} \mathbb{E}_{z \sim p(z|x_n)}[-\log q(y_n|z)] \\ + \beta \mathrm{KL}[p(Z|x_n), r(Z)] \tag{36}$$

where $N$ is the batch size, $x_n$ and $y_n$ are the input and output variables, $z$ is the latent representation, $p(z|x_n)$ is the encoder distribution, $q(y_n|z)$ is the decoder distribution, $r(Z)$ is a prior distribution (usually a standard Gaussian), $\beta$ is a Lagrange multiplier that controls the trade-off, and KL is the Kullback-Leibler divergence.

**NIB** Kolchinsky et al. (2019). Natural Information Bottleneck is a method that uses a natural gradient to optimize the IB Lagrangian with a linear entropy function[2]. The loss function of NIB is given by:

$$L_{NIB} = \frac{1}{N} \sum_{n=1}^{N} \mathbb{E}_{z \sim p(z|x_n)}[-\log q(y_n|z)] \\ + \beta H(Z|x_n) \tag{37}$$

where $H(Z|x_n)$ is the conditional entropy of $Z$ given $x_n$, and the other symbols are the same as in VIB.

**VIB-squared** Rodríguez Gálvez et al. (2020). Squared Variational Information is a variant of VIB that uses a quadratic entropy function instead of a linear one[3]. The loss function of squared-VIB is given by:

$$L_{\text{VIB-squared}} = \frac{1}{N} \sum_{n=1}^{N} \mathbb{E}_{z \sim p(z|x_n)}[-\log q(y_n|z)] \\ + \beta H^2(Z|x_n) \tag{38}$$

where $H^2(Z|x_n)$ is the squared conditional entropy of $Z$ given $x_n$, and the other symbols are the same as in VIB.

**NIB-squared** Rodríguez Gálvez et al. (2020). Squared Natural Information Bottleneck is a variant of NIB that uses a quadratic entropy function instead of a linear one. The loss function of squared-NIB is given by:

$$L_{\text{NIB-squared}} = \frac{1}{N} \sum_{n=1}^{N} \mathbb{E}_{z \sim p(z|x_n)}[-\log q(y_n|z)] \\ + \beta H^2(Z|x_n) \tag{39}$$

where $H^2(Z|x_n)$ is the squared conditional entropy of $Z$ given $x_n$, and the other symbols are the same as in NIB.

## E    OTHER EXPERIMENTAL RESULTS

**Parameter Sensitivity of Conventional IBs.** We provide additional experimental details, as shown in Tables 3 and 4, presenting the results of experiments with other Information Bottleneck (IB) methods under a fixed $\beta$ value. This is to illustrate the sensitivity of traditional IB methods to hyperparameters.

**Further Evidence of the Superiority of MIB.** We show in Figures 6 and 7 a comparison of the adversarial attack and generalization performance of different methods during the training process, to demonstrate the superiority of our approach.

Table 3: Comparison of the performance of different information bottleneck methods on Fashion-MINST with fixed $\beta = 0.01$.

| Attack | VIB | NIB | VIB-squared | NIB-squared |
|---|---|---|---|---|
| FGSM Goodfellow et al. (2014) | 46.16±4.37 | 53.36±5.00 | 50.36±0.59 | 59.47±1.00 |
| PGD Mądry et al. (2017) | 38.85±5.20 | 39.00±2.30 | 42.95±1.86 | 45.16±0.00 |
| NIFGSM Lin et al. (2019) | 58.05±2.98 | 60.98±4.14 | 61.83±0.49 | 66.76±1.26 |
| EOTPGD Liu et al. (2018) | 38.26±5.16 | 38.86±2.43 | 42.25±1.71 | 44.73±0.25 |
| MIFGSM Dong et al. (2018) | 28.34±6.84 | 33.56±6.62 | 35.02±3.50 | 44.52±2.91 |
| UPGD Mądry et al. (2017) | 26.47±6.88 | 28.19±5.13 | 33.35±3.64 | 38.97±3.41 |
| Jitter Schwinn et al. (2023) | 41.98±3.84 | 44.60±2.48 | 46.33±1.42 | 50.38±0.69 |

Table 4: Comparison of the performance of different information bottleneck methods on CIFAR-10 with fixed $\beta = 0.01$.

| Attack | VIB | NIB | VIB-squared | NIB-squared |
|---|---|---|---|---|
| FGSM Goodfellow et al. (2014) | 26.90±2.15 | 42.79±1.90 | 33.55±5.71 | 44.81±0.85 |
| PGD Mądry et al. (2017) | 27.14±1.67 | 41.18±2.55 | 32.81±5.19 | 42.83±1.13 |
| NIFGSM Lin et al. (2019) | 30.96±1.90 | 53.07±0.81 | 39.28±8.63 | 54.71±1.26 |
| EOTPGD Liu et al. (2018) | 27.03±1.87 | 41.29±2.21 | 33.07±4.77 | 42.68±1.20 |
| MIFGSM Dong et al. (2018) | 26.00±2.02 | 39.17±2.82 | 31.41±4.08 | 41.10±1.28 |
| UPGD Mądry et al. (2017) | 25.76±2.29 | 39.56±2.97 | 31.51±4.32 | 41.24±1.42 |
| Jitter Schwinn et al. (2023) | 25.65±2.19 | 43.04±1.71 | 32.85±5.62 | 45.24±1.16 |

# F  RELATED WORK FOR MULTI-OBJECTIVE OPTIMIZATION

## F.1  MULTI-OBJECTIVE OPTIMIZATION

In the context of multi-objective optimization (MOP), the goal is to optimize multiple conflicting objectives simultaneously. Formally, a MOP can be defined as follows:

$$\min_{\mathbf{x} \in \mathcal{X}} \{f_1(\mathbf{x}), \dots, f_m(\mathbf{x})\} \tag{40}$$

where $\mathcal{X} \subseteq \mathbb{R}^n$ represents the decision space, and $f_i : \mathcal{X} \to \mathbb{R}$ for $i = 1, \dots, m$ are $m$ objective functions, each to be minimized. In such problems, it is typically impossible to find a single solution that optimizes all objectives simultaneously because the objectives often conflict with one another. Instead, the concept of *Pareto optimality* is used to describe solutions that strike a balance between the different objectives Miettinen (1999).

**Definition 1** (Pareto Optimality). *A solution $\mathbf{x}^* \in \mathcal{X}$ is called Pareto optimal if there does not exist another $\mathbf{x} \in \mathcal{X}$ such that $f_i(\mathbf{x}) \leq f_i(\mathbf{x}^*)$ for all $i = 1, \dots, m$, and $f_j(\mathbf{x}) < f_j(\mathbf{x}^*)$ for at least one $j$.*

In other words, a solution is Pareto optimal if no other solution can improve one objective without worsening at least one other objective. The set of all Pareto optimal solutions is called the *Pareto set*, and its corresponding image in the objective space is referred to as the *Pareto front*. The goal in MOPs is often to approximate the Pareto front as well as possible.

## F.2  EXISTING APPROACHES FOR MULTI-OBJECTIVE OPTIMIZATION

Numerous methods have been proposed to solve MOPs, each offering different strategies for approximating the Pareto front and handling the trade-offs between competing objectives. Below, we review some of the key approaches and contrast them with the method adopted in this paper.

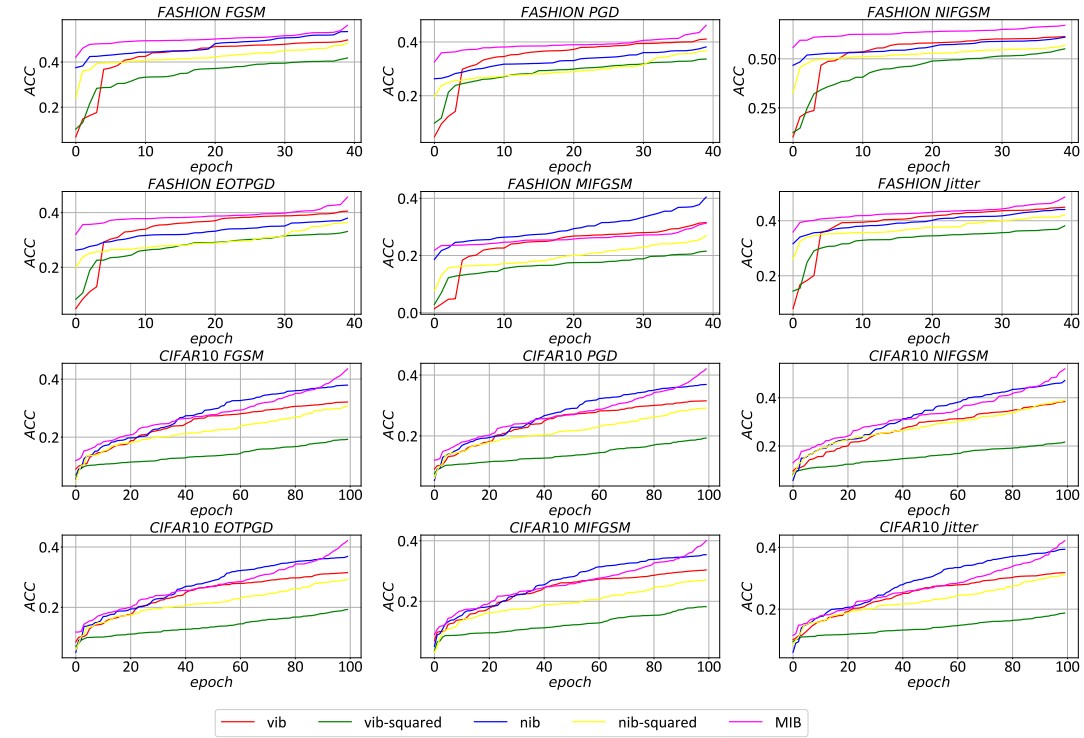

Figure 6: Comparison of the dynamic adversarial performance of different information bottleneck methods.

**1. Scalarization Methods**    Scalarization methods, such as the *weighted sum method* and the $\epsilon$-*constraint method*, transform the multi-objective problem into a series of single-objective optimization problems. In the weighted sum method, a set of scalar weights $\alpha_i$ is assigned to each objective, and the scalarized objective function is formulated as:

$$\min_{\mathbf{x} \in \mathcal{X}} \sum_{i=1}^{m} \alpha_i f_i(\mathbf{x}) \tag{41}$$

While this method is simple and effective for convex problems, it has several limitations. For instance, it can struggle to approximate non-convex Pareto fronts Miettinen (1999). Moreover, the choice of weights $\alpha_i$ requires careful tuning, as it directly affects the trade-offs between objectives. This is a key limitation compared to the method proposed in this paper, which *automatically determines the optimal weight combination* using the *min-norm solution* to balance the objectives dynamically, without requiring manual weight selection.

**2. Pareto-based Evolutionary Algorithms**    Evolutionary algorithms such as *NSGA-II* (Non-dominated Sorting Genetic Algorithm II) Deb et al. (2002) and *MOEA/D* (Multi-Objective Evolutionary Algorithm based on Decomposition) Zhang & Li (2007) have been widely used to solve MOPs. NSGA-II employs a non-dominated sorting mechanism to identify Pareto optimal solutions and uses a crowding distance metric to maintain diversity along the Pareto front. MOEA/D decomposes the MOP into scalar subproblems and solves them in parallel, which helps in approximating the entire Pareto front.

These evolutionary methods are particularly effective in exploring large, complex search spaces and approximating diverse Pareto fronts. However, they can be computationally expensive and slow to converge, especially for problems with high-dimensional decision spaces. In contrast, the *min-norm solution* approach presented in this paper is more computationally efficient, as it focuses on finding

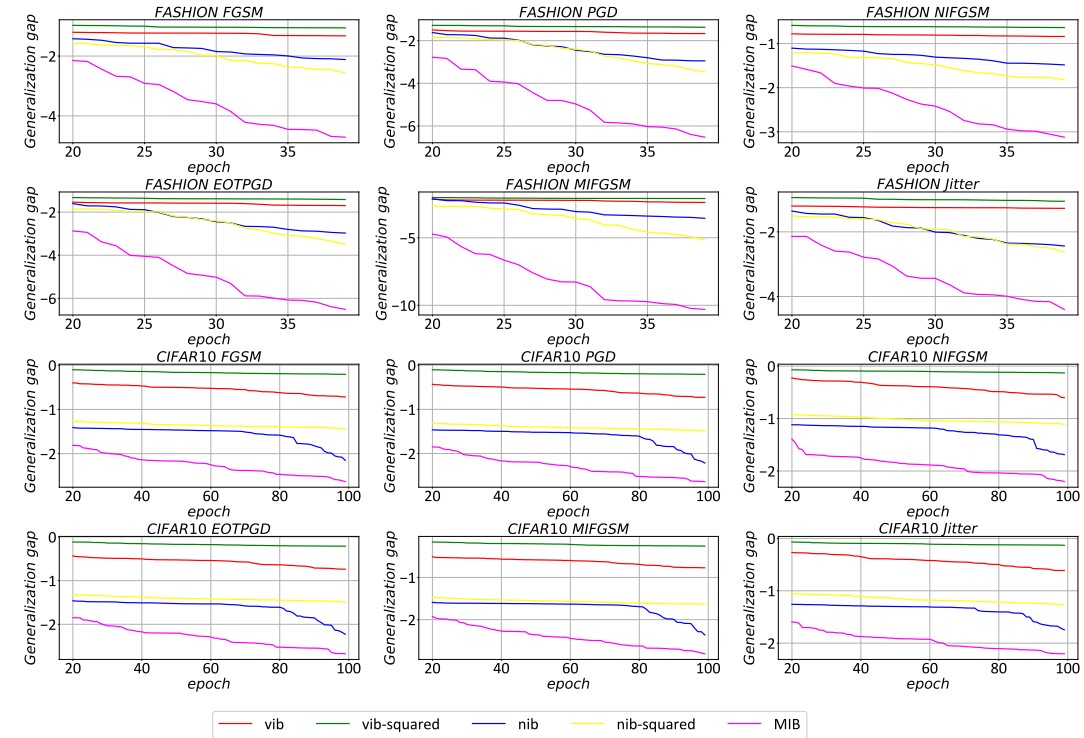

Figure 7: Comparison of generalization gap during the training process of different information bottleneck methods.

a single optimal solution by minimizing the norm of the gradient combination rather than evolving a population of solutions over multiple generations.

**3. Gradient-Based Multi-Objective Optimization** Gradient-based techniques, such as the *Multi-Gradient Descent Algorithm (MGDA)* Desideri (2012), optimize multiple objectives by combining their gradients through linear scalarization or other methods. MGDA aims to find a direction that simultaneously improves all objectives, and the update is performed in that direction. The method proposed in this paper builds on this idea but introduces a more sophisticated approach by finding the *min-norm combination of gradients*. Unlike MGDA, our method is specifically tailored for the information bottleneck (IB) problem, and thanks to the non-linear weight allocation and smoothing mechanisms, it demonstrates superior capability in finding solutions on the non-linear frontiers.

# G ALGORITHM DESCRIPTION

In this section, we present the pseudocode description of the core algorithm of this paper.

**Algorithm 1** MIB Training Process

**Require:** Model parameters $\theta$, batch data $B$, learning rate $\eta$, temperature parameter $\mu$, nonlinear weight parameter sqrted

**Ensure:** Updated model parameters $\theta$

1: Get input $x$ and label $y$ from batch data $B$
2: Forward propagation: $z = f_\theta(x)$  $\qquad\qquad\qquad\qquad$ ▷ $f_\theta$ is the neural network model
3: Calculate prediction loss: $\mathcal{L}\text{pred} = \text{CrossEntropy}(z, y)$
4: Calculate information loss: $\mathcal{L}\text{info} = \text{MutualInformation}(z, x)$
5: Calculate gradients:
6: $\qquad$ gpred $= \nabla\theta\mathcal{L}\text{pred}$
7: $\qquad$ ginfo $= \nabla_\theta\mathcal{L}_{\text{info}}$
8: Apply Frank-Wolfe algorithm to solve for optimal weights:
9: $\qquad \mathbf{w}^* = \text{FrankWolfe}(\mathbf{g}_{\text{pred}}, \mathbf{g}_{\text{info}})$
10: Apply nonlinear weight transformation:
11: **for** $i \in$ pred, info **do**
12: $\qquad \tilde{w}_i = (w_i^*)^{1+\sqrt{\text{sqrted}}}$
13: **end for**
14: Calculate minimum loss: $\mathcal{L}\text{min} = \min(\tilde{w}\text{pred}\mathcal{L}\text{pred}, \tilde{w}\text{info}\mathcal{L}_{\text{info}})$
15: Construct Smooth-IB loss function:
16: $\qquad \mathcal{L}\text{base} = \tilde{w}\text{pred}\mathcal{L}\text{pred} + \tilde{w}\text{info}\mathcal{L}\text{info}$
17: $\qquad \delta\text{pred} = (\tilde{w}\text{pred}\mathcal{L}\text{pred} - \mathcal{L}\text{min})/\mu$
18: $\qquad \delta\text{info} = (\tilde{w}\text{info}\mathcal{L}\text{info} - \mathcal{L}\text{min})/\mu$
19: $\qquad \mathcal{L}\text{smooth} = \mu\log(\exp(\delta_{\text{pred}}) + \exp(\delta_{\text{info}}))$
20: $\qquad \mathcal{L}\text{Smooth-IB} = \mathcal{L}\text{base} + \mathcal{L}\text{smooth} + \mathcal{L}\text{min}$
21: Calculate gradient of Smooth-IB loss:
22: **for** $i \in$ pred, info **do**
23: $\qquad \alpha_i = \frac{\exp((\tilde{w}i\mathcal{L}i - \mathcal{L}\text{min})/\mu)}{\sum k\in\text{pred,info}\exp((\tilde{w}k\mathcal{L}k - \mathcal{L}\text{min})/\mu)}$
24: **end for**
25: $\nabla\theta\mathcal{L}\text{Smooth-IB} = \tilde{w}\text{pred}\nabla_\theta\mathcal{L}\text{pred} + \tilde{w}\text{info}\nabla_\theta\mathcal{L}\text{info}$
26: $\nabla\theta\mathcal{L}\text{Smooth-IB}+ = \alpha\text{pred}\tilde{w}\text{pred}\nabla\theta\mathcal{L}\text{pred} + \alpha\text{info}\tilde{w}\text{info}\nabla\theta\mathcal{L}_{\text{info}}$
27: Update model parameters: $\theta \leftarrow \theta - \eta\nabla_\theta\mathcal{L}_{\text{Smooth-IB}}$
$\qquad\qquad$ **return** Updated model parameters $\theta$

