# OpenReview forum: "Exploring Complex Trade-offs in Information Bottleneck through Multi-Objective Optimization"
_ICLR.cc/2025/Conference — Submitted to ICLR 2025_

### Official Review · Reviewer_ZH9G · 2024-10-24

**Soundness:** 1
**Presentation:** 1
**Contribution:** 2
**Rating:** 1
**Confidence:** 3

**Summary:**

The authors address the problem of tunning the Lagrange multiplier during optimization of DNNs with variational IB approximating loss functions.
A novel interpretation of the information curve as a Pareto-Optimal frontier is suggested, and a weighted linear programing approach to find this frontier is established, presumably converging to an optimal point on the Pareto-Optimal frontier.
Empirical experiments follow that demonstrate the utility of the new method in comparison to other variational IB methods.

**Strengths:**

* Novel approach to IB*
The paper suggests a novel interpretation of the IB information curve as a Pareto-Optimal frontier between rate and distortion. This is a novel and interesting concept that can inspire more research, for IB optimization or for IB application in fields such as game theory and economics.

* Experiments*
Experiments are conducted over different flavors of variational IB, and over many different types of adversarial attacks, showing promising results in almost all metrics.

**Weaknesses:**

* Technical correctness*
While the concept of the information curve as a Parteo-Optimal frontier is very interesting, it has been shown in the original IB paper [1] that the information curve adheres to the same Pareto-Optimal definitions. Hence, to the best of my understanding, this is not a new definition, rather a new interpretation. Moreover, the paper assumes that the Lagrangian formulation of the IB problem is a relaxation, where in fact it is not. The IB is a private case of Rate-Distortion [2] where distortion is measured by MI between compressed representation and a downstream variable. It's formulation $min I(X;Z)$ s.t. $I(Z;Y)\ge D>0$ is equivalent to the Lagrangian $I(X;Z)-\beta I(Z;Y)$. Provided the above, we have that no single point on the information curve, or the Pareto-Optimal frontier, is objectively superior to another point on the curve. Tradeoffs between rate and distortion are task specific, and currently have no formal hierarchy established in the general case.
In addition, I am not an expert on Linear Programing, but to my understanding the Frank-Wolfe algorithm requires that the function it optimizes will be strictly convex, and the solution domain to be constrained. While the information curve itself is convex [1] we have from [1] that: "The functional $\( F \left[ p(\tilde{x} | x); p(\tilde{x}); p(y | \tilde{x}) \right] \)$ is convex in each of the distributions independently but not in the product space of these distributions.". Meaning that the decoder objective is not necessarily convex, and cannot be optimized using the FW algorithm.
Another caveat is the existence of a trivial solution to IB. Following the data processing inequality, and the IB Markov chain $Y-X-Z$, we have that $I(X;Z) \ge I(Z;Y)$, hence any optimization with $\beta \le 1$ can yield a trivial nullifying solution, that will yield maximal compression and maximal distortion, further substantiating the need to tune $\beta$.
Finally, MIB proposes to drop the hyperparameter $ \beta$, but introduces a new hyperparameter $ \mu$ in the smoothing term.

* Rigor of experiments*
The provided wxperiment indeed show promise, and the multitude of objectives and attack methods used are impressive. But yet, the datasets used are, in my opinion, of to low of a dimension to justify strong empirical evidence. This is especially true because, to my understanding, the proposed method is a heuristic and not a new theorem, and that the original VIB paper showed experiments over IMAGENET, which is a substantially harder problem than FMNIST and CIFAR10.

* Novel findings*
The proposed interpretation of the information curve as a Pareto-Optimal frontier, together with the proposed LP approach to optimize it, are novel and interesting.

* Clarity of submission*
I've found it unclear in the following ways:
1. The main claim of the paper was not clear enough to me:
    a. Is the proposed method a new theorem claiming that one point on the information curve is objectively superior to other? If so, a theorem is required.
    b. Is the proposed method a heuristic that converges to a generally balanced rate-distortion tradeoff? If so, this needs to be stated, and such a balance needs to be defined. Also, a theorem to why it is preferred, or at least intuition is also required.
2. The motivation to use Frank-Wolfe algorithm instead of gradient descent is unclear.
3. The Frank-Wolfe variant used is not clearly explained, what are the weights and to what do we expect them to converge? Why do we weight the gradients?
4. I've found many places in the paper to be incoherent, or inaccurate, for example:
    (a) The intro features some very general claims that are, in my opinion, not coherent and not precise. in lines 39-40: "Latent representations are features that capture fundamental and hidden information about the data". The use of the term 'information' here is confusing and to my understanding wrong. The correct phrasing might be hidden features or hidden distributions in some cases. Also, the claim that latent representations allow us to 'ignore' irrelevant data is also very general and imprecise, DNNs can learn extremely an overfitted latent space.
    (b) Citation style does not follow guidelines and is confusing [Please see ICLR guidelines Section 4.1].
    (c) Many sentences are grammatically incorrect, such as this one in lines 58-59: " the Information Bottleneck method has been
applied with success in various deep learning tasks due to the theoretical insights of the Information Bottleneck framework and the variational techniques implemented by it". This sentence implies that the IB method itself has implemented techniques.
    (d) In lines 109-110: "ensures that we can find the trade-off between multiple objectives in the optimal optimization direction": What is "finding the tradeoff", this is both grammatically confusing, and unclear as to how we define optimality.
    (e) In Lines 286-287: "This gradient form ensures that all objectives receive appropriate attention during the optimization
process while providing a smoother optimization surface". Appropriate attention is ill defined, a more rigorous definition needs to be used.

* Citations*
[1] Naftali Tishby, Fernando C. Pereira, and William Bialek. The information bottleneck method. In The 37th annual Allerton Conference on Communication, Control, and Computing, Hebrew University, Jerusalem 91904, Israel, 1999.
[2] Claude E. Shannon. Coding theorems for a discrete source with a fidelity criterion. In IRE National Convention, 1959.

**Questions:**

I ask the authors to address the weaknesses clause, and also to provide the following clarifications:
1. Assuming we converge to a point on the Pareto-Optimal frontier, how can we claim this point is better than any other? How is this different than simply choosing any beta value?
2. To my understanding the Frank-Wolfe algorithm has two requirments: (a) That the optimized function be strictly convex. (b) A constraint on the space of possible solutions. Am I correct? If so, can we still use it to optimize the proposed loss given that the decoder is not convex over $\phi , \theta$? What are the constraints we put on the solution set and how do we chose them?
3. Please clarify the flavor of Frank-Wolfe used, why do we use it instead of gradient descent, and why do we put weights on the gradient?

It is possible that I didn't completely understand the proposed optimization, if this is the case following the clarifications, I am willing to increase my score.

---

### Official Review · Reviewer_ctem · 2024-11-02

**Soundness:** 2
**Presentation:** 2
**Contribution:** 2
**Rating:** 3
**Confidence:** 5

**Summary:**

The paper proposes an algorithm that adaptively determines the trade-off weights between compression and prediction. The method is demonstrated to automatically find Pareto-optimal solutions, achieving a better performance than the variational information bottleneck and its variant.

**Strengths:**

The paper applies multi-objective optimization to address the trade-off information bottleneck. Its results show better performance on adversarial attacks.

**Weaknesses:**

Overall, the paper is not novel enough, the motivation is unclear, and the experiments are not promising enough. Additionally, some references are missing.

### Novelty:

1. Applying the Frank-Wolfe method directly to an information bottleneck is not novel. Frank-Wolfe's method is well investigated.

2. The paper only explores the variational Information bottleneck. However, there are many different ways to estimate MI. I'm not sure if it still works on other types of IB. e.g., HSIC-IB[1], CS-IB[2].



### Motivation:

1. Figures 1 and 2 don't demonstrate the motivation well. The paper claims that the choice of Lagrange multiplier significantly affects the model's performance and robustness. I see some sensitivity, but is it possible for different MI estimators that are not stable and accurate enough to lead to unstable performance? Searching for the optimal parameters is easy now since there are many hyperparameter search tools, such as Optuna and WandB.


### Experiments:

1. Only test the method on cifar10 and fashion MNIST. Could you please add more experiments on Large-scale datasets like ImageNet?
2. Could you also test the method with different IB methods, like HSIC-IB, CS-IB, and NIB?
3. Additionally, test the method for out-of-distribution tasks, like domain generalization.
4. Disentangled IB [3] is another related work; please compare it with your proposed method.
5. Please visualize the information plane and verify that your method can achieve maximum compression.

### Missing references:

[1] Ma, Wan-Duo Kurt, et al. "The HSIC bottleneck: Deep learning without back-propagation." Proceedings of the AAAI conference on artificial intelligence. Vol. 34. No. 04. 2020.

[2] Yu, Shujian, et al. "Cauchy-Schwarz Divergence Information Bottleneck for Regression." The Twelfth International Conference on Learning Representations.

[3] Pan, Ziqi, et al. "Disentangled information bottleneck." Proceedings of the AAAI Conference on Artificial Intelligence. Vol. 35. No. 10. 2021.

**Questions:**

1. Please provide some references and a more detailed description of the smooth information bottleneck.

---

### Official Review · Reviewer_DMsE · 2024-11-03

**Soundness:** 2
**Presentation:** 3
**Contribution:** 3
**Rating:** 5
**Confidence:** 3

**Summary:**

This paper redefines the Information Bottleneck (IB) problem as a multi-objective optimization problem, eliminating the need for manually tuning the Lagrange multiplier to balance compression and prediction. The authors also demonstrate that MIB achieves more stable and optimal trade-offs.

**Strengths:**

The paper is very well-structured, clearly explaining the rationale for introducing a multi-objective approach. It also provides a proof of the Pareto Optimality of the proposed method. Additionally, the authors conducted extensive experiments to demonstrate the effectiveness of this approach.

**Weaknesses:**

In Eq (8), the constraint on the reweighted parameter $w$ is simply $\sum_i w_i=1$。 Based on my understanding, the smaller $g_i$ is, the larger $w_i$ will become, which suggests that many of the coefficients $w_i$ could end up being zero.

Additionally, during training, it seems that each time the loss function is computed, an optimization algorithm is executed, which could potentially slow down the algorithm.

Theoretically, I think further discussion might be necessary. For example, when the number of classes approaches infinity, will this weighting approach converge to the average? Moreover, the oracle property of these adaptive weights may also need further exploration.

From the writing perspective, I suggest rephrasing Eq (8)–Eq (12) in the form of an algorithm.

**Questions:**

I have some concerns about Figure 4. Which dataset does this figure refer to? The t-SNE results show that MIB separates the results well, suggesting a high accuracy, but in Tables 1 and 2, MIB does not show a significant improvement compared to VIB and NIB.

Did the authors consider testing on datasets beyond CIFAR-100, MNIST, or CIFAR? How would MIB perform when there are many classification categories or different data patterns?

If my questions are well-addressed, I'll be happy to raise my score.

---

### Official Review · Reviewer_GenM · 2024-11-08

**Soundness:** 3
**Presentation:** 3
**Contribution:** 2
**Rating:** 5
**Confidence:** 2

**Summary:**

In the traditional IB approach, Lagrange multipliers are used to balance the trade-off between compression and prediction, typically represented by a parameter $\beta$. This approach avoid choosing the $\beta$ and transforms the problem into a multiple objective optimization problems, both the compression (minimizing mutual information between the input and latent representation) and prediction (maximizing mutual information between the latent representation and the target).

**Strengths:**

1. This approach replaces the manual tuning of Lagrange multipliers with an automatic exploration of the Pareto frontier, allowing for more nuanced trade-offs.
2. It includes comparisons with traditional IB methods and demonstrates superior robustness, stability, and performance under adversarial attacks.

**Weaknesses:**

1. To improve the stability of the optimization process, the paper adopts the Smooth Information Bottleneck (Smooth-IB) technique following the MIB method. Notably, other IB methods like Variational IB (VIB) and Noisy IB (NIB) do not require this additional smoothing. This raises questions about the inherent stability of the MIB objective and whether it may be less stable than other methods. Additionally, implementing Smooth-IB may add to the overall complexity of the approach.

2. The experimental validation is primarily conducted on relatively small image datasets, such as FashionMNIST and CIFAR-10. This raises concerns about the scalability of the MIB approach to larger or higher-dimensional datasets, such as ImageNet, or more complex models. It would be valuable to assess whether there is a practical upper limit where MIB’s performance or stability declines, and if so, how this could impact its broader applicability.

**Questions:**

1. Can MIB be used without Smooth-IB?

2. Can MIB be extend to more complex models and datasets?

**Details Of Ethics Concerns:**

No ethics concerns

---

### Meta-Review · Area_Chair_TekF · 2024-12-22

**Metareview:**

The paper proposes a gradient-based multi-objective optimization algorithm that adaptively determines the weights to avoid the need of tuning the Lagrange multiplier of the variational IB loss functions. While the paper has provided comprehensive experimental results, the following questions remain unanswered after the rebuttal stage (no rebuttal was given) and I think they are important:
1. Experiments are done on small datasets and no experiments on larger datasets like ImageNet has been given to further verify the effectiveness of the method. The proposed method is more like a heuristic than theory, so it is important to have larger-scale experiments.
2. The method needs a smoothing technique which is not required by other IB methods like VIB and NIB, which raises questions about the stability of the proposed method.
3. The speed can potentially be slow.

Therefore I recommend rejection.

**Additional Comments On Reviewer Discussion:**

No rebuttal was given. The reviews have been summarized above.

---

### Decision · Program_Chairs · 2025-01-22

Reject